# Multi-Drug Resistant Pathogenic *Escherichia coli* Isolated from Wild Birds, Chicken, and the Environment in Malaysia

**DOI:** 10.3390/antibiotics11101275

**Published:** 2022-09-20

**Authors:** Mohamed-Yousif Ibrahim Mohamed, Jalila Abu, Zunita Zakaria, Abdul Rashid Khan, Saleha Abdul Aziz, Asinamai Athliamai Bitrus, Ihab Habib

**Affiliations:** 1Department of Veterinary Pathology and Microbiology, Faculty of Veterinary Medicine, Universiti Putra Malaysia, Serdang 43400, Selangor, Malaysia; 2Veterinary Public Health Research Laboratory, Department of Veterinary Medicine, College of Agriculture and Veterinary Medicine, United Arab Emirates University, Al Ain P.O. Box 1555, United Arab Emirates; 3Department of Veterinary Clinical Studies, Faculty of Veterinary Medicine, Universiti Putra Malaysia, Serdang 43400, Selangor, Malaysia; 4Department of Public Health and Medicine, Penang Medical College, George Town 10450, Penang, Malaysia; 5Department of Veterinary Microbiology and Pathology, Faculty of Veterinary Medicine, University of Jos, Jos P.M.B 2084, Nigeria; 6Department of Environmental Health, High Institute of Public Health, Alexandria University, Alexandria 21511, Egypt

**Keywords:** pathogenic *Escherichia coli*, MDR, wild birds, chickens, environment, Malaysia

## Abstract

Transmission of pathogenic microorganisms in the last decades has been considered a significant health hazard and pathogenic *E. coli*, particularly antibiotic-resistant strains, have long been identified as a zoonotic problem. This study aimed to investigate multidrug resistant pathogenic *E. coli* isolates from wild birds, chickens, and environment in selected Orang Asli and Malay villages in Peninsular Malaysia. The bacteriological culture-based technique, disc diffusion method, and multiplex Polymerase Chain Reaction (mPCR) assay was used to determine the occurrence of pathogenic *E. coli* strains in the several samples in the study. *E. coli* isolates showed a variety of multi-drug resistant (MDR) antibiotypes and Enteropathogenic *E. coli* (EPEC) and Enteroinvasive *E. coli* (EIEC) were the most predominantly identified pathogenic *E. coli* strains. The findings of this study demonstrated the significance of animal reservoirs and the environment as sources of pathogenic *E. coli,* resistant bacteria, and resistance genes. Hence, there is a need for adoption of a practical surveillance approach on MDR pathogens to control foodborne contamination.

## 1. Introduction

Antimicrobial resistance (AMR) is currently a major problem worldwide that threatens ecosystem health. If left unchecked, it is predicted that by 2050, a greater number of cases of human fatalities, severe economic losses, and a significant decrease in livestock production will occur [1,2]. Consequently, antimicrobial-resistant bacteria from poultry and other food animal sources have been on the rise worldwide [3]. These changes in resistance can be attributed to several factors, including the use of antimicrobial agents as feed additives, antimicrobial use as growth promoters, and overuse of antimicrobial agents in human and veterinary medicine [4]. A major concern to public health is the emergence of multidrug-resistant (MDR) foodborne pathogens [4,5]. The definition of MDR is acquired resistance to at least one antibiotic in three or more antibiotic classes. Strains of *E. coli* exhibiting MDR are considered as the most significant challenge in food safety. The dissemination of multidrug-resistant *E. coli* is one of the biggest threats to global health. Of significant importance, MDR pathogenic *E. coli* is the major cause of nosocomial infections, which are associated with high morbidity, case fatality, and increased healthcare costs [6,7].

The interaction among wild birds, chickens, humans, and their household environments can catalyze the sharing of resistance and virulence genes [8]. Several studies have reported that in communities where poultry farming is common, the households’ soil is contaminated with antibiotic residues from animals and humans, thus leading to an increase in the spread and dissemination of resistance determinants because of environmental contamination [1,9,10]. Other studies have reported the importance of wild birds in the spread of resistant bacteria and resistance and virulence genes to humans, chickens, farms, and the environment. Hence, the interaction between humans, wild birds, and other domestic animals is of public health concern, since it has the potential to accentuate life-threatening illnesses that can be difficult to treat [11,12,13].

*Escherichia coli* is a bacterium with a unique place in the microbial world, since it can not only cause life-threatening illnesses in animals and humans, but it also represents a large proportion of the autochthonous microbiota of different hosts [14]. The organism has high adaptive capacity, a feat that allows *E. coli* to survive for long periods of no growth and in a variety of ecological niches. This is partly due to an array of virulence genes acquired via horizontal transmission of pathogenicity, plasmids, bacteriophages, and transposons [15]. Generally, the pathogenic *E. coli* is broadly classified into two major categories, diarrheagenic or intestinal *E. coli* and extraintestinal *E. coli*. Based on the epidemiological and clinical features, specific virulence factors, and other characteristics, which include enterotoxin production and adherence phenotypes, six different pathogenic classes of diarrheagenic *E. coli* have been identified, namely, enteropathogenic *E. coli* (EPEC), enteroaggregative *E. coli* (EAEC), enterohemorrhagic *E. coli* (EHEC)—also known as Shiga toxin-producing *E. coli* (STEC)—enteroinvasive *E. coli* (EIEC), enterotoxigenic *E. coli* (ETEC), and diffuse-adhering *E.*
*coli* (DAEC) [16,17].

EPEC is among the most important foodborne pathogens worldwide [15]. EPEC expresses the eae protein depending on the presence or absence of the bundle-forming pilus A gene (*bfpA*), and can be classified into typical EPEC (tEPEC) and atypical EPEC (aEPEC). EPEC is well-recognized pathogen in developing countries; humans are generally considered a reservoir for tEPEC, while aEPEC is reportedly more prevalent in developing and developed countries, and animals are a major reservoir hosts [18,19]. EAEC is a major cause of acute and persistent diarrhea in children and adults globally, while ETEC is reported to be an emerging cause of foodborne diseases in Asia, Europe, and the USA [18]. EHEC/STEC causes bloody diarrhea (hemorrhagic colitis), non-bloody diarrhea, and hemolytic uremic syndrome (HUS). It is an important cause of foodborne infections in the USA, mainly due to contaminated meat and cattle, identified as being major reservoirs. Subsequently, a wide variety of food items are associated with disease, including sausages, unpasteurized milk, lettuce, cantaloupe melon, apple juice, and radish sprouts. EHEC has also caused numerous outbreaks associated with recreational and municipal drinking water, person-to-person transmission, and petting zoo and farm visitations. EHEC strains of the O157:H7 serotype are the most important EHEC pathogens in North America, the United Kingdom, and Japan, but several other serotypes, particularly those of the O26 and O111 serogroups, can also cause disease and are more prominent than O157:H7 in many countries [20]. EIEC is a major source of infection in humans, as no animal reservoirs have been identified, and it is reported to be common in low-income countries, where poor general hygiene favors its fecal–oral transmission [21].

ETEC causes watery diarrhea, which can range from mild self-limiting disease to severe purging disease. The organism is an important cause of childhood diarrhea in developing countries and is the main cause of diarrhea in travelers to developing countries [20]. In several studies, DAEC has been implicated as a cause of diarrhea, particularly in children >12 months of age; one study indicated that DAEC infection could be pro-inflammatory and that this effect can potentially be important in the induction of inflammatory bowel disease [20].

Several cases of foodborne diseases (reported as food poisoning) have been reported in Malaysia, of which pathogenic *E. coli* could be one of the most plausible causes; however, in most reports, the available data did not link specific organisms to reported cases of food poisoning. The education of food handlers in improving their standards of hygiene is essential to reduce the risk of foodborne illnesses, diseases, and poisoning.

Sanches et al. [22] reported that free-living wild birds, chickens, and humans in villages were act as carriers of EPEC and EIEC in Europe, Japan, and the USA. Hence, this study aimed to investigate multidrug-resistant pathogenic *E. coli* from wild birds, chickens, humans, and environmental samples in some Orang Asli and Malay villages in Peninsular Malaysia.

## 2. Results

### 2.1. Multidrug Resistant E. coli Isolates

All *E. coli* isolates showed resistance to all antibiotics tested, with 100% MDR in *E. coli* from wild birds in Orang Asli villages and 44.4% in *E. coli* from wild birds in Malay villages. MDR ranged from 15 to 100%, in *E. coli* isolates from chickens. *Escherichia coli* isolates in chickens from village (F) showed 100% MDR as shown in Table 1.

### 2.2. Occurrence of E. coli Virulence Genes in Wild Birds, Chickens, and Environment in Villages 

The overall occurrence of *eae*A genes of *E. coli* isolates recovered from wild birds, chickens, and environmental samples were 48/196 (24.5%). This consisted of 7 (12.1%) from wild birds, 38 (45.2%) from chickens, and 16 (27.6%) from the environment (Table 2, Table 3 and Table 4). Among wild birds, *eae*A was predominantly identified in isolates recovered from the Eurasian Tree Sparrow 6 (42.9%) and the white-Vented Myna 1 (100%), all of them being from the Malay villages, and none of the wild birds from Orang Asli villages had any of the studied virulence genes. The number of *E. coli* isolates recovered from chickens in Orang Asli village were 22 (55%) and Malay villages 16 (36.4%), which carried the *eaeA* gene. Additionally, the modified mPCR revealed the presence of EPEC 6 (7.1%) and EIEC 2 (2.4%) in chickens (Figure 1 and Figure 2). Of the 54 *E. coli* isolates isolated from the environment, 3 (16.7%) from flies, 6 (33.3%) from water, and 7 (38.9%) from soil samples were found to carry the *eae*A gene. A modified mPCR assay showed that none of the isolates from flies were EPEC and EIEC. Similarly, only 1 (11.1%) of the isolates recovered from water and soil were EPEC and EIEC, respectively, being from Orang Asli and Malay Villages, respectively.

Table 5 shows the occurrence of *eae*A gene in *E. coli* isolates from wild birds, two of them being resistant to four antibiotics (both isolated from village D), four wild bird isolates which were resistant to three antibiotics (three from village D and one from village F), and one resistant to one antibiotic (isolated from village F). EPEC and EIEC were not detected in the wild birds. 

For the chicken *E. coli* isolates, the *eae*A gene was detected in two isolates resistant to seven antibiotics (both isolated from village F), two isolates resistant to six antibiotics (from villages C and F), four isolates resistant to five antibiotics (two isolated from village B, two from village D, and one from village E), two isolates resistant to four antibiotics (isolated from villages C and E), four isolates resistant to three antibiotics (one isolated from village B, one from village C, and two from village E), thirteen isolates resistant to two antibiotics (nine isolated from village C, one from village D, and three from village E), and ten isolates resistant to one antibiotic (seven isolated from village C, two from village D, and one from village E). The EPEC were identified in one chicken isolate, which was resistant to seven antibiotics (isolated from village F), one isolate resistant to six antibiotics (isolated from village F), three isolates resistant to two antibiotics (one isolated from village C and the other two isolated from village E), and one isolate resistant to one antibiotic (isolated from village D). The EIEC were identified in two isolates which were resistant to two antibiotics (one isolated from village C and E each).

The *eae*A gene was detected in one environmental isolate which was resistant to eight antibiotics (isolated from village E), one isolate resistant to seven antibiotics (isolated from village E), five isolates resistant to six antibiotics (four isolated from village B and one from village C), one isolate resistant to five antibiotics (isolated from village B), one isolate resistant to four antibiotics (isolated from village C), four isolates resistant to three antibiotics (two isolated from village D, one from village C, and one from village E), and three isolates resistant to two antibiotics (two isolated from village A and one from village D) (Table 5). The EPEC was detected in one isolate which were resistant to six antibiotics (isolated from village B). The EIEC was detected in one environmental isolate resistant to seven antibiotics (isolated from village E).

### 2.3. Statistical Analysis

From the analysis there were significant differences in the occurrence of MDR *E. coli* between the two locations (Orang Asli villages and Malay villages) in wild birds (*p* < 0.0001) and chickens (*p* = 0.0423) Table 6.

## 3. Discussion

The present study revealed the presence of the eae gene in 12.1% of the wild bird *E. coli* isolates from the Malay villages, 71.4% of them being identified as MDR. A study on wild birds in Japan [19] reported an occurrence of the *eae**A* gene as 25% with high MDR, and indicated that wild birds are a reservoir of atypical enteropathogenic *E. coli* (EPEC) and antibiotic resistance genes. According to a study in the UK by Hughes et al. [23], although wild birds are unlikely to be direct sources of STEC strains, they do represent a potential reservoir of other virulent genes. This, coupled with their ability to act as long-distance vectors of STEC, means that wild birds have the potential to influence the spread and evolution of pathogenic *E. coli* groups, such as EPEC and EHEC. In a study in Tunisia by Yahia et al. [24], the occurrence of *eae**A* in wild birds was found to be low at 8.3%. The *eae**A* gene could occur in several groups of pathogenic *E. coli*, such as STEC and EHEC [20,21]. Thus, the occurrence of this gene indicates that there are other possible pathogenic *E. coli* groups in the birds in Malay villages. In Malaysia, no published data are available on the prevalence of the *eae**A* gene, EPEC, and EIEC or their antibiotic resistance patterns in wild birds, chickens, and humans. The few published data reveal the presence of pathogenic *E. coli* strains and their resistance to multiple antibiotics. Most studies have concentrated on beef and poultry samples [7,8].

The high-level occurrence of the MDR *E. coli eae**A* gene in the wild birds in the Malay villages could be caused by several factors, among which are the environmental factors associated with the feeding habits of these birds. Different feeding habits influence the presence of pathogenic *E. coli* in wild birds, as reported in some surveys [23,25]. This could be because these groups of birds in Malay villages feed on human garbage [24] that was probably contaminated by bacteria carrying the *eaeA* and antibiotic resistance genes. Moreover, these birds might become infected with the *eaeA* and antibiotic resistance genes from animal farms, as suggested in some studies [8,9,10,11,12,13,14,15,16,17,18,19,20,21,22,23,24,25,26], or from chickens in these villages, as has been observed. Chickens have been identified as commonly carrying EPEC and EIEC. Several wild birds (e.g., Eurasian tree sparrow) have been seen inside chicken houses, sharing the feed with chickens.

The different locations of wild birds (Orang Asli villages and Malay villages) were a significant factor for the occurrence of MDR *E. coli* in these birds. Isolates from wild birds in the Orang Asli villages and Malay villages showed 100% and 44.4% MDR, respectively. This high MDR *E. coli* in wild birds in Orang Asli villages probably because these birds could have fed on human garbage and vegetation that were likely to be contaminated with *E. coli* with high resistance. Additionally, the high humidity [27] and low temperature [28] in Orang Asli villages is a suitable atmosphere for the *E. coli* to survive. The occurrence of antibiotic resistance genes in a Mediterranean river and their persistence in the riverbed sediment has been reported [27]. The wild birds in the Orang Asli villages could have acquired the resistance genes from the river or from the riverbed sediment, whereas there was no river near the Malay villages. In a study on the impact of river water on the community of tetracycline-resistant bacteria and the structure of tetracycline resistance genes [28], it was noted that the bacteria of the genera Aeromonas sp. and Acinetobacter sp. were able to transfer 6 out of 13 tested tet genes into *E. coli*, which can promote the spread of antibiotic resistance in the environment.

Chicken isolates showed a high prevalence of the *eae**A* gene, at 45.2%, and 39.5% of them were MDR. The occurrence of EPEC was 7.1%, and 33.3% of them showed MDR, while EIEC was 2.4% with no MDR. The occurrence of the *eae**A* gene in the Orang Asli village chicken samples was found to be high, at 55%, with 27.3% of them showing MDR, while those in the Malay villages was at 36.4%, and 56.3% of the samples showed MDR. Enteropathogenic *E. coli* (EPEC) in the Malay village chicken samples was found to be high, at 11.4%, with 40% MDR, while those in Malay villages was at 2.4%, and the EIEC in chicken isolates was found to be positive in two isolates: one in an Orang Asli village (2.5%) and the other in a Malay village (2.3%). Other studies have shown that the prevalence of the *eaeA* gene in chicken isolates varies from low to high. A high contamination rate has been reported in Japan (62.6%) [25], while a lack of contamination has been found in Brazil (0%) [29] and in France (0%) [30]. According to a study in Sao Paulo, Brazil [29] on isolates of atypical enteropathogenic *E. coli* (EPEC) from chickens and chicken-derived products, the results indicate that chicken and chicken products are important sources of atypical enteropathogenic *E. coli* (EPEC) strains that could be associated with human disease, highlighting the need to improve hygiene practices in chicken slaughtering and meat handling processes.

There was a higher occurrence of MDR *E. coli* in the isolates from the chickens from the Orang Asli villages at 70.5% than in those from the Malay villages at 47.5%. The different locations of chickens were a significant factor for the occurrence of MDR *E. coli* in these chickens. The high rate of MDR *E. coli* present in chickens from the Orang Asli villages is possibly because the chickens were exposed to contaminated environment, as they were released every day from morning to evening to roam and feed in the open environment as well as on human wastes, and the wild birds might also play a significant role in contaminating the environment of the Orang Asli villages. However, in the Malay villages, although the chickens were kept all day in houses which were of open type, the high occurrence of MDR *E. coli* was because these chickens were most likely exposed to contaminated water and to wild birds and pests such as flies and other insects that freely entered the houses.

The high rates of the MDR *E. coli eae**A* in the chicken isolates in Malay villages may be due to the poor hygiene in the chicken houses or because the chickens acquired the *E. coli* pathogenic *eae**A* and resistance genes from the environment. This study detected the MDR *E. coli*
*eaeA* from the environment in villages D and E. The high occurrence of the *eae**A* gene in the chicken isolates in these villages (31.3% and 53.3%, respectively) might be due to environmental factors, especially the soil and water. Moreover, wild birds might have a role in the occurrence of the MDR *E. coli eaeA* in the chickens in the Malay villages in this study, as 27.8% of the wild bird isolates tested positive in these villages. In this study, the Eurasian tree sparrow showed high MDR *E. coli eaeA* in village D (100%), but none in village F (0%). It was observed that in village D, several Eurasian tree sparrows gained access to the chicken houses and shed their droppings in them, thereby contaminating the floor, feed, and water. Thus, Eurasian tree sparrows could play a significant role in village D in terms of the occurrence of the *eae**A* gene in chickens.

This study showed the prevalence of the *eae**A* gene, EPEC, and EIEC as being 29.6%, 1.9%, and 1.9% in the environment, respectively; however, the isolates from village F were negative for the *eae**A* gene, EPEC, and EIEC. The occurrence of the *eae**A* gene in *E. coli* isolates in the environment in the Orang Asli villages was found to be higher, at 37%, and 80% of them showed MDR compared to that of the Malay villages at 22.2%, of which 66.7% showed MDR. This high prevalence of the MDR-*eae*A-*E. coli* in the environment of the Orang Asli villages might be because in this study it was observed that during the day the chickens were released to the open environment to scavenge for food; thus, they might shed the MDR-*eae*A-*E. coli* through feces, spreading them in the soil and water in the villages. In Malay villages, the chickens were kept in their houses with almost no access to contaminate the environment compared to the chickens in the Orang Asli villages. Thus, the chickens do play an important role in contaminating the environment in Orang Asli villages.

## 4. Materials and Methods

### 4.1. Sample Collection

Wild birds: The location of the trap in each village identified for capturing birds were among the houses in the villages or not more than 5 km away from the villages. In each location, a trap (mist net) was set up and placed for six hours. This was undertaken in the morning. Every twenty minutes, the trap was checked for birds. A photograph of the bird was taken for identification and each bird was marked by a red band around one of its legs to avoid being resampled. A cloacal swab was taken before the bird was released.

Healthy humans, chickens, and the environment: fresh stools from humans, cloacal swabs from chickens, and samples the from environment, including the soil, drinking water, and flies, were collected. The locations of the villages in Perak and Kedah are shown in Figure 3.

### 4.2. Confirmation of E. coli Isolates

A total of 196 *E. coli* isolates were obtained from the previous studies by Mohamed et al. [7] and Mohamed et al. [31]. All the isolates were isolated from wild birds (*n* = 58), chickens (*n* = 84), and environment (flies, water, and soil) (*n* = 54). The *E. coli* isolates were confirmed using routine bacteriological culture and PCR assay. Briefly, *E. coli* isolates were recovered by culture on Brilliance *E. coli*/Coliform Selective Media (Oxoid). A single colony from each positive culture plate was collected and used for this study.

### 4.3. Detection of eaeA, EPEC, and EIEC Using Monoplex and Modified Multiplex PCR Assay

#### 4.3.1. Genomic DNA Extraction

Genomic DNA was extracted using boiling method as described by Kamaruzzaman et al. [32]. Briefly, a suspension of overnight *E. coli* fresh culture was prepared in a 1.5 mL microcentrifuge tube (Eppendorf, Australia) containing 100 µL sterile distilled water. The cell suspension was incubated at 94 °C for 10 minutes in a dry heat block and then allowed to cool down to room temperature. The suspension was then centrifuged at 13, 000× *g* for 5 minutes. The supernatant was then transferred into a new 1.5 mL microcentrifuge and used as a DNA template for PCR assay. 

#### 4.3.2. Primer and PCR Cycling Conditions Modified Multiplex PCR to Detect EPEC and EIEC

Detection of *aea*A, *ial* and *bfpA* genes, *eae*, SHIG, and *bfp*A primers were used for the identification of EPEC and EIEC was purchased from Next Gene Scientific Sdn Bhd (Table 7). Modified multiplex PCR to detect EPEC and EIEC was carried out in a 50 µL PCR mixture, which encompassed 200 ng (equivalent of 5 µL) of bacterial DNA extract, 10 µM of primer mix (1 µL each), 14 µL of RNase free from water, and 5U/µL (25 µL) of m-PCR master mix 2x (Qiagen). The PCR was optimized using known EPEC (ATCC 43887) and EIEC (ATCC 43893) for the positive controls. The bacterial strain *E. coli* (ATCC 11775) DNA extract was replaced with the equivalent amount of sterile distilled water for the negative control (Table 7). The PCR amplification procedure was performed as described by Nguyen et al. [33]. The initial activation step was at 96 ℃ for 4 minutes, followed by 30 cycles at 94 °C for 20 seconds, 55 °C for 20 seconds and an extension of 72 °C for 10 s. This was performed in the VeritiTM 96-Well Eppendorf Thermal Cycler. Similar cycling conditions were performed for the conventional PCR to detect the *eaeA* gene in the *E. coli* isolates. Amplicons were resolved in 2% agarose gel (Agarose, LE Analytical Grade) prepared using 1x Tris-Borate-EDTA buffer (2 mM EDTA, 40 mM Tris-Borate, PH 7.5). Then, 3 µL/mL Gel-red stain was mixed with the PCR products and run for 90 min at 75 V. The electrophoresed gel was viewed using a gel documentation system under transilluminator UV light.

### 4.4. Antibiotic Susceptibility Test

All the 196 *E. coli* isolates recovered from wild birds, chickens, and environment were tested against ten panel of antibiotics representing eight different categories (Table 8). The ten antimicrobial agents include streptomycin (10 µg), gentamicin (10 µg), tetracycline (30 µg), ciprofloxacin (5 µg), enrofloxacin (5 µg), nalidixic acid (30 µg), ampicillin-sulfabactam (10 µg), sulphamethoxazole-trimethoprim (25 µg), erythromycin (15 µg), and cefpodoxime (10 µg). Antimicrobial susceptibility testing was performed using the disc diffusion method; the diameter of each inhibition zone was measured and interpreted according to the guidelines and recommendation of Clinical Laboratory Standard Institute [34]. *Escherichia coli* ATCC 25922 and *Pseudomonas aeruginosa* ATCC 27853 were used as quality control strains.

### 4.5. Statistical Analysis

Data for the occurrence of MDR-*E. coli* in wild birds and chickens from different locations were analyzed by Chi square test. The statistical significance was considered at *p* < 0.05.

## 5. Conclusions

The obtained results revealed Enteropathogenic *E. coli* (EPEC) and Enteroinvasive *E. coli* (EIEC) as the most predominant isolates circulating among wild birds, chickens, and the environment. A variety of MDR antibiotypes were also observed, this evidenced the roles of wild birds, chickens, and the environment as sources of transmission of Antimicrobial resistant bacteria and resistance genes via the food value chain. This underscores the need to develop surveillance strategies and control procedures to reduce the use of antibiotics, and subsequently, the development of antimicrobial resistance.

## Figures and Tables

**Figure 1 antibiotics-11-01275-f001:**
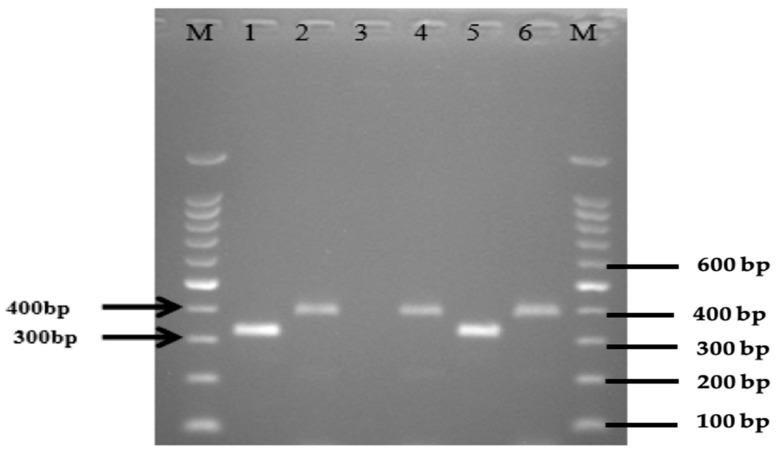
Modified multiplex PCR on representative EPEC and EIEC in *E. coli* isolates isolated from chickens. Lane M: marker 100 bp ladder, lane 1: EIEC ATCC 43893 as positive control, lane 2: EPEC ATCC 43887 as positive control, lane 3: *E. coli* ATCC 11775 as negative control, lanes 4, 6: EPEC; lane 5: EIEC.

**Figure 2 antibiotics-11-01275-f002:**
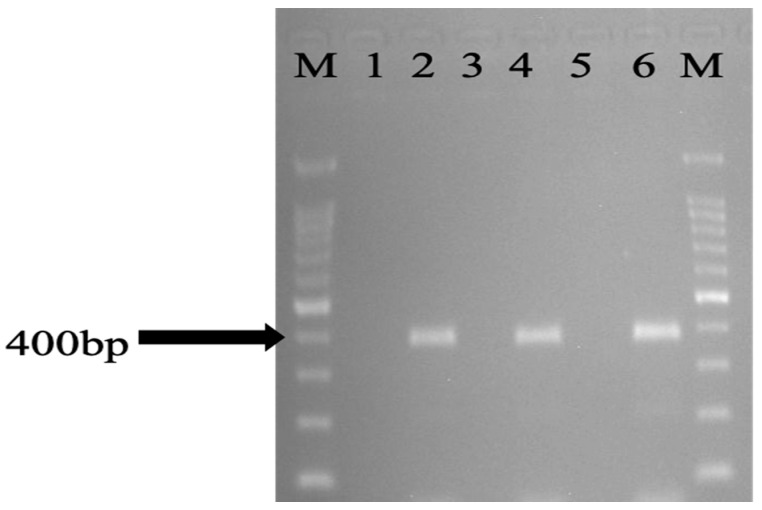
PCR amplification of representative *eaeA* gene in *E. coli* isolates isolated from the environment. Lane M: marker 100 bp ladder, lane 6: *eaeA* ATCC 43887 as positive control, lane 5: *E. coli* ATCC 11775 as negative control, lanes 2, 4: *eaeA*, lanes 1, 3: negative.

**Figure 3 antibiotics-11-01275-f003:**
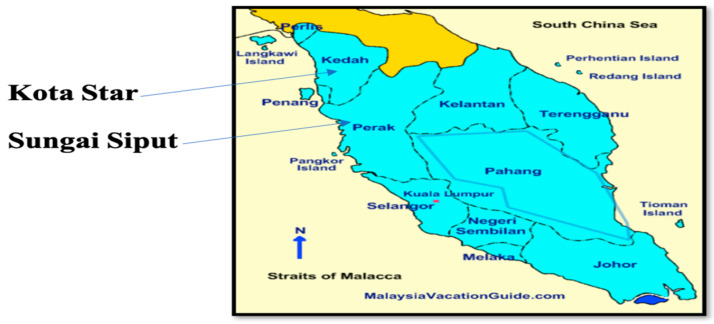
Locations of villages in Perak and Kedah.

**Table 1 antibiotics-11-01275-t001:** Multidrug resistant *E. coli* isolates in wild birds and chickens according to the village.

Villages	No. of Isolates	No. (%) Resistant Isolates	No. of Antibiotics Resistant to *	No. (%) MDR
Wild birds				
A	9	9 (100%)	4–8	9 (100%)
B	21	21 (100%)	4–7	21 (100%)
C	10	10 (100%)	4–8	10 (100%)
D	9	9 (100%)	1–4	6 (66.7%)
E	3	3 (100%)	1	0 (0%)
F	6	6 (100%)	1–3	2 (33.3%)
	58	5 (100%)	1–8	48 (82.8%)
Chickens				
A	10	10 (100%)	2–9	8 (80%)
B	10	10 (100%)	1–8	8 (80%)
C	20	20 (100%)	1–6	3 (15%)
D	16	16 (100%)	1–5	10 (62.5%)
E	15	15 (100%)	1–5	8 (53.3%)
F	13	13 (100%)	3–8	13 (100%)
	84	84 (100%)	1–9	50 (59.5%)

Note: * range in the number of antibiotics that the isolates were resistant to; MDR = resistant to at least one antibiotic in three or more classes.

**Table 2 antibiotics-11-01275-t002:** Detection of *eaeA* gene in *E. coli* isolates from wild birds in the studied villages from Orang Asli and Malay (*n* = 58).

Village	Wild Bird Species	No. of Isolates	No. *eaeA* Gene Positive (%)
A	Oriental Magpie Robin	2	0 (0)
	White-rumped Shama	4	0 (0)
	Little Spiderhunter	3	0 (0)
B	Oriental Magpie Robin	13	0 (0)
	White-rumped Shama	8	0 (0)
C	Oriental Magpie Robin	9	0 (0)
	Little Spiderhunter	1	0 (0)
D	Eurasian Tree Sparrow	8	5 (62.5)
	White-Vented Myna	1	1 (100)
E	Eurasian Tree Sparrow	2	0 (0)
	Jungle Myna	1	0 (0)
F	Eurasian Tree Sparrow	4	1 (25)
	White-Vented Myna	0	0 (0)
	Jungle Myna	2	0(0)
Total		58	7 (12.1%)

**Table 3 antibiotics-11-01275-t003:** Number of *E. coli* isolates (EPEC and EIEC), and detection of the *eae* gene by conventional PCR in the chicken samples from the studied villages.

Village	Chicken Isolates	*eaeA* Gene	EPEC	EIEC
A *	10	1 (10%)	0 (0%)	0 (0%)
B *	10	3 (30%)	0 (0%)	0 (0%)
C *	20	18 (90%)	1 (5%)	1 (5%)
Total no.	40	22 (55%)	1 (2.5%)	1 (2.5%)
D #	16	5 (31.3%)	1 (6.3%)	0 (0%)
E #	15	8 (53.3%)	2 (13.3%)	1 (6.7%)
F #	13	3 (23.1%)	2 (15.4%)	0 (0%)
Total no.	44	16 (36.4%)	5 (11.4%)	1 (2.3%)
Total	84	38 (45.2%)	6 (7.1%)	2 (2.4%)

* Orang Asli villages, Sungai Siput, Perak; # Malay villages Kota Setar, Kedah.

**Table 4 antibiotics-11-01275-t004:** Detection of the eaeA gene by conventional PCR and the bfpA and iac genes by multiplex-PCR, for the identification of the EPEC and EIEC isolates, respectively, in the environmental samples.

Village	Flies (Three Isolates Per Village)			Water (Three Isolates Per Village)			Soil (Three Isolates Per Village)			Total		
	*eae*A Gene (%)	EPEC (%)	EIEC (%)	*eae*A Gene (%)	EPEC (%)	EIEC (%)	*eae*A gene (%)	EPEC (%)	EIEC (%)	*eae*A Gene (%)	EPEC (%)	EIEC (%)
A	0 (0)	0 (0)	0 (0)	2 (66.7)	0 (0)	0 (0)	0 (0)	0 (0)	0 (0)	2 (22.2)	0 (0)	0 (0)
B	1 (33.3)	0 (0)	0 (0)	2 (66.7)	1 (33.3)	0 (0)	2 (66.7)	0 (0)	0 (0)	5 (55.6)	1 (11.1)	0 (0)
C	1 (33.3)	0 (0)	0 (0)	0 (0)	0 (0)	0 (0)	2 (66.7)	0 (0)	0 (0)	3 (33.3)	0 (0)	0 (0)
Total *	2 (22.2)	0 (0)	0 (0)	4 (44.4)	1 (11.1)	0 (0)	4 (44.4)	0 (0)	0 (0)	10 (37)	1 (3.7)	0 (0)
D	0 (0)	0 (0)	0 (0)	1 (33.3)	0 (0)	0 (0)	2 (66.7)	0 (0)	0 (0)	3 (33.3)	0 (0)	0 (0)
E	1 (33.3)	0 (0)	0 (0)	1 (33.3)	0 (0)	0 (0)	1 (33.3)	0 (0)	1 (33.3)	3 (33.3)	0 (0)	1 (11.1)
F	0 (0)	0 (0)	0 (0)	0 (0)	0 (0)	0 (0)	0 (0)	0 (0)	0 (0)	0 (0)	0 (0)	0 (0)
Total #	1 (11.1)	0 (0)	0 (0)	2 (22.2)	0 (0)	0 (0)	3 (33.3)	0 (0)	0 (0)	6 (22.2)	0 (0)	1 (3.7)
Total	3 (16.7)	0 (0)	0 (0)	6 (33.3)	1 (5.6)	0 (0)	7 (38.8)	0 (0)	0 (0)	16 (29.6)	1 (1.9)	1 (1.9)

* Orang Asli villages Sungai Siput, Perak; # Malay villages Kota Setar, Kedah.

**Table 5 antibiotics-11-01275-t005:** Carriage of *eae*A gene, EPEC, EIEC, and antibiotypes of *E. coli* from wild birds, chickens, and tenvironment.

Sample ID	Antibiotype	No. Ab	*eae*A Gene	EPEC	EIEC		
**AS4 *, FW2**	**** ESxtCipCpdSEnrTeCnSamNa**	10	-	-	-		
AC7	ESxtCpdSEnrTeCnSamNa	9					
AF13	ESxtCipSEnrTeCnSamNa						
FF5, FW18	ESxtCipCpdSEnrTeCnNa						
FS8	ESxtCipCpdSEnrTeSamNa						
AWb6	ESxtCpdSEnrTeCnNa	8	EW15	-	-		
CWb36	ECipCpdSEnrTeSamNa						
BC7	ESxtCpdSEnrTeSamNa						
FC20, EW15, FW8	ESxtCipSEnrTeCnNa						
AF8, EW2, EW17, FS3, FS9	ESxtCipCpdSEnrTeNa						
AF9	ESxtCipSEnrTeSamNa						
AWb1	ECipCpdSEnrTeNa	7	FC11, FC14, ES1	FC14	ES1		
AWb9, BWb10, BWb29	ESxtSEnrTeSamNa						
FC11, FC14, FC25, ES13	ESxtCipSEnrTeNa						
FC15, FF4, FF6	ESxtCipEnrTeCnNa						
FC28	ESxtSEnrTeCnSam						
AS6	ESxtCpdSTeSamNa						
AS10, ES1, ES5	ESxtCpdSEnrTeNa						
AWb5, CWb31, CWb35	ECipSEnrTeNa	6	CC12, FC1, BF7, BW2, BW6, BS4, CS11	FC1, BW6			
BWb13	ECpdSEnrTeNa						
BWb15, BWb16, BWb21, BWb30, BF7, BW2, BW6, BS4	ESxtSEnrTeSam						
BWb27, CWb34, FC1, DF28	ESxtSEnrTeNa						
AC4	ESxtCpdSTeSam						
CC12	ESxtCipEnrTeNa						
CS11	ESxtCipSTeNa						
AWb3	ECipSTeNa	5	EC8, BC4, BC2, DC6, DC12, BS2 -				
AWb4, DF15, BW7	ESxtSEnrTe						
BWb11	ECipSEnrTe						
BWb14, BWb24, BWb25, FC21	ESxtEnrTeSam						
BWb20, BWb26, BWb28, CWb37, CWb38, FC18, BS2	ESxtSTeSam						
CWb33, BC8, BF9	ESEnrTeNa						
AC2, AC10	ESxtCpdSTe						
BC2, EC8	ECipEnrTeNa						
BC4	ESxtCpdTeSam						
DC6, DC12, EC7	ESxtEnrTeNa						
FC9, DW9	ESxtSTeNa						
EF1	ESxtSTeCn						
CW1	ESxtCpdSSam						
BS9	EEnrTeSamNa						
AWb2, AWb8, BWb12, BWb18, EC11	ESxtEnrTe	4	DWb2, DWb3, CC13, EC11, CF11				
AWb7, BWb17, CWb32, CWb39, CWb40, DWb2, DWb3, BC5, BC9, CC13, DC1, DC3, DC5, DC9, CF11, DF2	ESxtSTe						
BWb18	ESEnrTe						
BWb22, BWb23	ESxtTeSam						
AC6	ECpdSTe						
BC6, BF2	EEnrTeNa						
DC7, FC8	ESxtTeNa						
FWb32, BC1, EC18, FC29, CF5, DS19	ESxtTe	3	DWb1, DWb4, DWb5, BC1, EC18, CC14, EC12, DS19, EF2, DW11, CS13				
DWb4, DWbK5, DWb6, DC2, DC11, DC13, EC3, EC5, EF2, EF3, DW11, DW12, CS12, CS13	ESTe						
DWb1, FWb22, CF4	ETeSam						
AC3	ECpdTe						
AC5	ECpdS						
AC8	ESxtS						
CC14, EC12, FC13	ETeNa						
EC14	SxtCpdS						
FWb21, FWb29, AC1, BC11, CC2, CC3, CC5, CC6, CC11, CC15, CC16, CC20, DC14, DC15, DC18, EC1, EC2, EC4, EC6, AW4, AW6, CW4, CW5, DS2, DS15	ETe	2	CC4, AC9, CC3, CC5, CC6, CC11, CC15, CC16, CC20, DC15, EC1, EC2, EC6, AW4, AW6, DS15	CC4, EC1, EC2,			CC6, EC6,
AC9	ECpd						
CC4	ENa						
DWb11, DWb12, DWb13, EWb14, EWb19, EWb20, FWb25, FWb35, BC3, CC1, CC7, CC8, CC9, CC10, CC17, CC18, CC19, DC17, DC19, DC21, EC9, EC13, EC17, AW1	E	1	DWb12, FWb35, CC1, CC7, CC8, CC9, CC10, CC17, CC18, DC17, DC19, EC9	DC17			

* Isolate ID: A: village, S: source (Wb: wild bird; C: chickens; F: flies; W: water; S: soil), 4: isolate number. ** Sam: ampicillin-sulfbactam, Te: tetracycline, Cn: gentamicin, E: erythromycin, Cip: ciprofloxacin, Na: nalidixic acid, Enr: enrofloxacin, Sxt: sulfamethoxazole-trimethoprim, Cpd: cefpodoxime, and S: streptomycin. No. Ab: number of antibiotics the isolates were resistant to.

**Table 6 antibiotics-11-01275-t006:** Occurrence of MDR *E. coli* (%) in wild birds and chickens.

Village	Wild Birds	Chickens
A *	100	80
B *	100	80
C *	100	15
D #	66.7	62.5
E #	0	53.3
F #	33.3	100
SEM	10.20621	31.78283
*p*. values	<0.0001 ^+^	0.0423 ^+^

* Orang Asli villages Sungai Siput, Perak. # Malay villages Kota Setar, Kedah. ^+^ Significant. SEM: Standard error of the mean (SEM).

**Table 7 antibiotics-11-01275-t007:** Oligonucleotide sequence for monoplex and modified multiplex PCR for the detection of *eaeA, bfpA*, and *ial*.

Primer	Target Gene	Oligonucleotide Sequence	Amplicon Size (bp)	Reference Strain	Category of Pathogenic *E. coli*
eae	*eaeA*	*FW: 5′CACACGAATAAACTGACTAAAATG-3′ RV: 5′AAAAACGCTGACCCGCACCTAAAT-3′	376	ATCC43887	*eaeA*
SHIG	*ial*	FW: 5′-CTGGTAGGTATGGTGAGG-3′ RV: 5′-CCAGGCCAACAATTATTTCC-3′	320	ATCC43893	EIEC
bfpA	*bfpA*	FW: 5′-TTCTTGGTGCTTGCGTGTCTTTT-3′ RV: 5′-TTTTGTTTGTTGTATCTTTGTAA-3′	367	ATCC43887	EPEC
				ATCC11775	Negative control

*FW: Forward, RV: Reverse.

**Table 8 antibiotics-11-01275-t008:** Antimicrobial class and clinical break points of antimicrobial agents tested against (*n* = 196) *E. coli* isolates recovered from wild birds, chickens, and environmental samples.

Antibiotic Class	Antimicrobial Agents	Disc Concentration (µg)	Clinical Break Points of Antimicrobial Agents (mm)		
			Susceptible	Intermediate	Resistance
Aminoglycosides	Streptomycin	10	≥15	12–14	≤11
	Gentamicin	10	≥15	13–15	≤12
Penicillin-combination	Ampicillin-sulfabactams	10	≥17	14–16	≤13
Tetracyclines	Tetracycline	30	≥19	15–18	≤14
Macrolides	Erythromycin	15	≥23	14–22	≤13
Quinolones	Nalidixic acid	30	≥19	14–18	≤13
Flouroquinolones	Enrofloxacin	5	≥21	18–20	≤17
	Ciprofloxacin	5	≥21	16–20	≤15
Cephalosporin/cephamycins	Cefpodoxime	10	≥21	18–20	≤17
Sulphamethoxazole-Trimethoprim	Sulpamethoxazole-trimethoprim	25	≥16	11–15	≤10

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
