# Peer review of "Multi-Drug Resistant Pathogenic Escherichia coli Isolated from Wild Birds, Chicken, and the Environment in Malaysia"

_antibiotics, 2022, doi:10.3390/antibiotics11101275_

Round 1

Reviewer 1 Report

Dear authors,

The manuscript fits well within the scope of the journal. You have investigated an interesting topic and the theme has been properly described. The objectives of the study were clearly defined. I would like to underline and applaud the authors for the novelty of their research as there are not many papers dealing with wild and domestic birds, as well as the environment.  The Introduction is written concisely and provides sufficient background. The methods have been properly described, and the design of the experiment and statistical methods applied to allow us to make reliable conclusions.

Results are well presented and thoroughly discussed and data interpretation is appropriate.

The manuscript is well written, presented and discussed, and understandable to a specialist readership.

No significant limitations have been detected, whereas the paper presents novel and useful findings. The results have significant practical implications. 

In conclusion, I recommend the acceptance of the manuscript for publication after minor corrections which are indicated within the attached file.

All the best

Author Response

Done the corrections. 

Thank you

Regards,

Reviewer 2 Report

The manuscript (antibiotics-1880117) submitted by Mohamed et al. consists of a study developed in several pre-selected villages from Asli and Malay villages in Malaysia, concerning the characterization of E. coli isolates obtained from animal and environmental samples.

I considered this study very relevant and important in the antibiotic resistance area, however, major alterations and revisions should be done by the authors, o that the manuscript can be accepted in MDPI-Antibiotics.

My first advice is that the English grammar and the manuscript punctuation should be corrected by an English native since in some parts is very difficult to follow. Also, the title is not appropriate to the presented work, I suggest something like: “Multi-drug resistant pathogenic Escherichia coli characterization isolated from wild birds, chicken, and environmental samples in some Orang Asli and Malay Villages from Peninsular Malaysia”.

Also, this study is a follow-up of two previous studies from the author, and probably due to that, some important information mainly in the material and methods is missing to better understand how and from where the E.coli isolates were obtained.

Now, concerning each section, my suggestions are:

Abstract:

All the abbreviations, like mPCR, EPEC, EIEC, and MDR must be in full.

Line 23: Please correct to “… have long been identified as a zoonotic problem.”

Line 26: Please correct to “… E. coli strains in the several samples in the study.”

Line 28: Please remove “in wild birds, chickens, and the environment”.

Introduction:

Several parts must be rewritten concerning the English quality.

Line 43/44 – antimicrobial “agents”, can be misunderstood with other microorganisms that are considered to have antimicrobial capacity, so please substitute by “antibiotics”.

Line 61/62: Correct “…concern since it has the potential to accentuate life-threatening illness that can be difficult to treat”.

Line 73: Remove “through” and “island”

Line 82: “Enteroaggregative E. coli” is repeated, please remove it.

Line 85: “EPEC expresses eae depending on the presence or absence of the bundle-forming pilus A gene” – expresses the eae protein? If so, remove the italic and add the word protein, please.

Line 111: Substitute “could” for “can”.

Line 121: Substitute for “… villages were pointed out to act as…”, and United States substitute by “USA”

Line 124: Substitute for “… chickens, humans, and environmental samples”

Results

Line 129: Substitute for “environmental samples”

Line 134/135: Substitute for “from the Eurasian Tree Sparrow and the white-Vented Myna species… being all of them from the Malay villages, since any of the Orang Asli villages had any of the studied virulence genes”.

Line 138: How many E. coli isolate were recovered from Malay Villages?

I presume that the punctuation of the sentence is wrong which can confuse the reader regarding the presented percentages for each village.

Line 143: Remove “each”

Line 144: Remove “both representing” and substitute for “being from”

Line 148 (figure 1): All legends (figures and tables) must be self-explanatory, so please introduce the scale in the marker lanes, the meaning of EPEC and EIEC, and their respective genes and amplicons sizes. Examples EPEC (bfpA gene – 367bp amplicon) and EIEC (iac gene – 320 bp amplicon).

Line 156 (table 1 legend): Correct the legend to: “Detection of eae gene in E. coli isolates from wild birds in the studied villages from Orang Asli and Malay”.

Also, I presume that A, B, C, D, E, and F are villages that were identified in a previous study of the team, however, is crucial in this study a map pointing out where the selected villages are localized in order to better understand the samples locations.

Line 159 (table 2 legend): substitute for “Number of E. coli isolates (EPEC and EIEC), and detection of the eae gene by conventional PCR in the chicken samples from the studied villages”.  

Once again what are the villages A, B, C D E and F?

Table 3 legend: Substitute for “Detection of the eae gene by conventional PCR and the bfpA and iac genes by multiplex-PCR, for the identification of the EPEC and EIEC isolates respectively, in the environmental samples”.

Page 7: The line numeration is missing and from this page onwards the pagination is also missing, along the line numbering has restarted in the discussion section. These errors make the correct revision of the manuscript very difficult.

Please correct: Table 4 shows the occurrence of eae A gene in E. coli isolates from wild birds, being two of them resistant to four antibiotics… and one resistant to one antibiotic…”

At the beginning of the next paragraph correct to: “For the chicken E. coli isolates the eae A gene was detected in two isolates resistant to seven antibiotics… two isolates resistant to six antibiotics…

Also, EPEC and EIEC are not “detected”, the respective genes are, so please substitute the word “detected” with “identified”.

Discussion

Once again, the English quality is very poor along with the punctuation, which makes it difficult to understand the beginning of the discussion.

Line 2: Correct to: “The present study revealed the presence eae gene in 12,1% of the wild birds E. coli isolates from the Malay villages, being 71,4% of them identified as MDR”.

Adapt this type of writing to the remaining sentences

Line 17 to 19: What are those “few studies”? Indicate the references.

Line 30 to 33: The idea in this sentence is already mentioned at the begging of the discussion, please remove this paragraph.

Line 34: The authors can not say chicken isolates, please correct along all the manuscript for “E. coli isolates from chicken samples” … or “E. coli isolates from the wild bird samples” or “E. coli isolates from the environmental samples”.

Line 44: “… low level in Brazil and France” with zero percent? That is not a low level it was not even detected! Again, very bad English.

Line 59: “… but low in village F (0%)”. 0% is zero not low.

Material and Methods:

Line 77 (Point 4.1): What is the meaning of this paragraph? This looks more like a result from a previous study, and something is missing to do the link to the present study.

Line 83 (Point 4.2): A map with the geographic location of the studied villages is crucial in this manuscript. How and from where were the E. coli isolates obtained? From feces of the animals? Blood? Or from another biological sample? This must be explained.

Line 93 (point 4.3): In my opinion, an PCR with only a pair of primers could be said as only “PCR” or conventional PCR, not monoplex.

Line 102: The multiplex PCR does not detect the EPEC or EIEC, it detects the bfp A and ial genes.

Line 104 to 108: More important than the volumes of each PCR reagent, is the concentration of each one of them, so please indicate what were the used concentrations. Also, the mPCR master mix is from which brand?

Line 111: correct to “table 5”.

Line 115: Correct to: “similar cycling conditions were performed for the conventional PCR to detect the eae gene in the E. coli isolates.

Table 5: I did not understand the reference strain in the negative control. Also, add the “Fw” and “Rv”, for forward and reverse primers respectively, in each pair of primers in the table. In the legend of the table, again, the conventional and mPCR do not detect EPEC and EIEC but do detect the respective genes, so please correct this.

Line 143 (point 4.4): Why did the authors choose this combination of antibiotics?

Line 145: Substitute “antimicrobial agents” for “antibiotics”.

Table 6 legend: Substitute for “… and environmental samples”. Also, the table is not formatted.

Conclusion

All this paragraph should be rewritten since it seems like a telegram, and the grammar is not corrected. Once again, an English native should review all the manuscript grammar and punctuation. The presented work is very interesting and important regarding the antibiotic resistance area, mainly in these types of countries where the available information is very scarce.  Also, the work would also be richer if an analogy was made with E. coli isolates obtained from human biological samples if this type of information is published or available. Otherwise, this could be a future challenge for the authors to consider.

Author Response

(The authors gave the same response as above.)

Reviewer 3 Report

1. 

Author Response

(The authors gave the same response as above.)

Round 2

Reviewer 2 Report

The authors assumed most of the suggestions and alterations made, however, I still have a problem with the English quality and with the presented map in the material and methods section. The map has to indicate which are the villages A, B ,C...

Also, in the material and methods section, the concentrations of the PCR reagents are still missing, 10% of primers and 10% of the mix do not represent the concentrations used. Please check in scientific articles on how to present this information. 

Author Response

The authors assumed most of the suggestions and alterations made, however, I still have a problem with the English quality and with the presented map in the material and methods section. The map has to indicate which are the villages A, B ,C...

English quality:

Done by MDPI Editing Services.  

The map:

The distant between the two kind of villages is very far.

Orang Asli villages in the mountain in Perak.

Malay villages in Kedah.

Also, we did not mention the names of the villages in the study.

Also, in the material and methods section, the concentrations of the PCR reagents are still missing, 10% of primers and 10% of the mix do not represent the concentrations used. Please check in scientific articles on how to present this information.

Response

Appropriate concentration of DNA template, primers used, and Master mix added.

Reviewer 3 Report

The authors made some effort in addressing some of the comments made by the reviewers. However, the authors failed to address some of the critical comments made by the reviewer. The manuscript must be improved in terms of presentation of the results and need to include statistical test to analyze the results. Specific comments listed below

1. Line 27: "EPEC and EIEC were the most predominantly identified pathogenic E. coli strains" Please include in which samples those species were commonly identified (ie., animal samples or environmental samples or both).

2. Line 73: It is not clear of the number of locations from which samples were collected. Please indicate it in the figure (figure 1) as well.

3. Please italicize "eae" gene in the Table3 caption

4. No statistical tests were used to analyze the results. A frequency table indicating the prevalence of different E. coli could be created and statistical test should be performed to identify samples with significant differences in isolation rates and percentage of multi drug resistance ( or statistical tests should be performed based on the study hypothesis)

5. The current representation of data in Table 4 is long and difficult to understand. A bar diagram comparing different samples or making separate table for different types of samples will be useful.

Author Response

The authors made some effort in addressing some of the comments made by the reviewers. However, the authors failed to address some of the critical comments made by the reviewer. The manuscript must be improved in terms of presentation of the results and need to include statistical test to analyze the results. Specific comments listed below

  1. Line 27: "EPEC and EIEC were the most predominantly identified pathogenic E. coli strains" Please include in which samples those species were commonly identified (ie., animal samples or environmental samples or both).

Added

Chickens have been identified as commonly carrying EPEC and EIEC.

  1. Line 73: It is not clear of the number of locations from which samples were collected. Please indicate it in the figure (figure 1) as well.

Added

location of the trap in each village

Wild birds: The location of the trap in each village identified for capturing birds were among the houses in the villages or not more than 5 km away from the villages. In each location, a trap (mist net) was set up and placed for six hours. This was undertaken in the morning. Every twenty minutes, the trap was checked for birds. A photograph of the bird was taken for identification and each bird was marked by a red band around one of its legs to avoid being resampled. A cloacal swab was taken before the bird was released.

  1. Please italicize "eae" gene in the Table3 caption

Done.

  1. No statistical tests were used to analyze the results. A frequency table indicating the prevalence of different E. coli could be created and statistical test should be performed to identify samples with significant differences in isolation rates and percentage of multi drug resistance (or statistical tests should be performed based on the study hypothesis)

Done

  1. The current representation of data in Table 4 is long and difficult to understand. A bar diagram comparing different samples or making separate table for different types of samples will be useful.

The authors acknowledge the feedback; however, we feel having different tables will amount to repetition as different samples exhibited similar antibiotypes.